# 3D Affordance Reconstruction from Egocentric Demonstration Video

## Abstract

Developing robots capable of generalized skills remains an exceedingly challenging task. Drawing from psychology, the concept of affordance has emerged as a promising intermediate representation to guide robot manipulation. However, prior work has primarily focused on 2D affordances from video, neglecting critical spatial information such as camera positioning, absolute position, depth and geometry. In this paper, we present a novel training-free method that constructs 3D affordances from egocentric demonstration videos. To address the challenge of insufficient static, high-quality frames for 3D reconstruction in egocentric videos, we employ the 3D foundational model DUST3R Wang et al. (2024), which reconstructs scenes from sparse images without requiring COLMAP Schönberger & Frahm (2016). We analyze videos using hand detection to identify contact times and 2D contact points, reconstruct these interactions using DUST3R, and project the 2D contact points into 3D space using gaussian heatmaps. Finally, we derive hand trajectories through 3D hand pose estimation and process them using linear regression to integrate the spatiotemporal dynamics of human-object interactions. We demonstrate the effectiveness of our method on the ego4d-exo dataset Grauman et al. (2024) for seven real-world hand-object manipulation tasks in cooking scenes.

## 1 Introduction

Teaching a home robot to generalize across daily tasks is highly challenging, particularly even straightforward manipulation tasks can rapidly escalate in complexity with variations in the environment. Although end-to-end reinforcement learning has seen significant advancements, achieving robust generalization remains challenging. In the domain of autonomous driving, intermediate representations such as monocular depth estimation, 3D object detection, and semantic segmentation have been developed to enhance system performance. Similarly, in the context of robot manipulation, various intermediate representations have been proposed to address these challenges and enhance task generalization.

The term "affordance," borrowed from psychology, was defined by James Gibson as 'an action possibility available in the environment to an individual, irrespective of their ability to perceive it.' In robotics, affordance typically refers to the ways objects can be utilized, and this concept is widely applied in the perception community to build representations that assist robots in learning how to interact with functional objects. These representations often take the form of heatmaps, sometimes complemented by usage trajectories, indicating how objects should be used.

Moreover, within the field of machine learning, there has been a predominant focus on extracting information from large-scale image and text datasets. However, in applications that involve interacting with the environment, such as robotics, there is a critical need for information beyond mere pixels or text. Effective robotic functioning requires spatial-temporal data. As such, we assert the importance of extracting information from videos, which provide rich content encompassing both spatial and temporal dimensions, as essential for advancing robotic capabilities.

While previous methods, as defined Bahl et al. (2023), view affordance primarily in terms of its contact point and the target object's post-contact trajectory in 2D pixel space from video, this approach introduces several problems in robotic learning. Firstly, depth perception remains ambiguous in pixel space, making it difficult for robots to ascertain the distance to objects. Furthermore, the

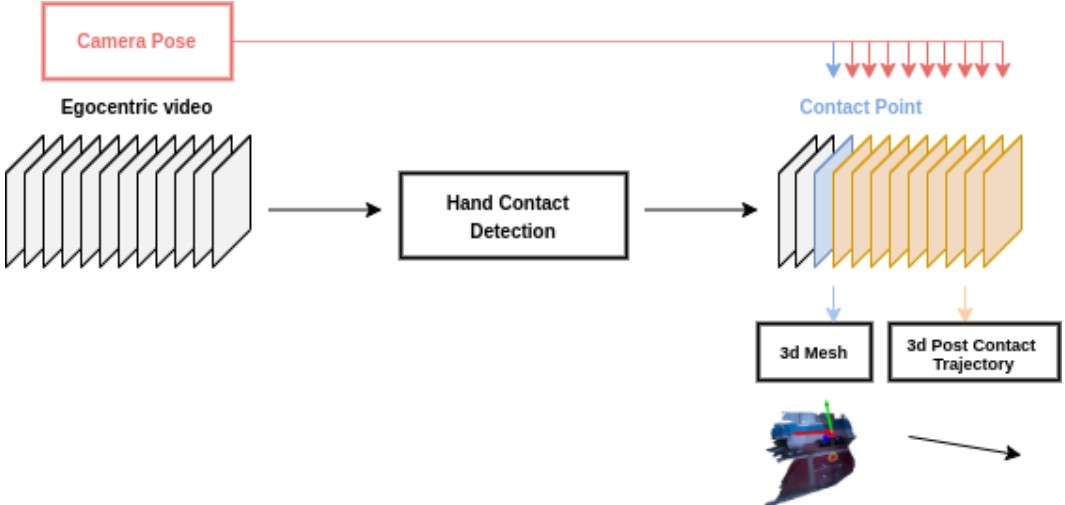

Figure 1: We propose a training-free method that can reconstruct 3D affordances from given egocentric videos. The core approach involves using hand detection to identify the contact frame. From there, we reconstruct the contact image and project the 2D contact point into a 3D. Finally, the 3D trajectory is obtained through linear regression of the sequence of 3D poses after contact.

geometry of objects cannot be accurately determined from 2D alone. As a result, robots struggle to comprehend the true scenario and accurately locate the absolute contact points. Lastly, neglecting to account for camera pose can lead to inaccuracies in locating the absolute positions of both the contact points and the object's subsequent 3D post-contact trajectory.

In this paper, we embrace the definition of affordance from VRB Bahl et al. (2023), which also includes the contact point and post-contact trajectory. We propose a method to reconstruct 3D affordance from egocentric video demonstrations of human interactions with objects. Our approach begins by detecting the contact time and point, followed by reconstructing the image using a 3D reconstruction foundation model to project the contact point onto the mesh. Lastly, we apply linear regression to the 3D pose estimated from the model, concatenating these elements to derive the 3D affordance. This includes a 3D mesh with a contact heatmap and the post-contact trajectory of the target object.

Our main contributions are as follows:

- We extend affordance learning from 2D to 3D space using egocentric video. Among these advancements, we can reconstruct the scene even when there is a lack of images revolving around it

- We propose a training-free method that quickly constructs 3D affordance without the need for COLMAP and depth

- We can reconstruct 3d affordance in real-world scenarios in ego4d-exo, where the egocentric video is of low quality and the egocentric camera, attached to a human, experiences significant movement and unstable viewpoints.

## 2 RELATED WORKS

### 2.1 AFFORDANCE

The concept of affordance, originally introduced by James J. Gibson in psychology, describes the potential actions enabled by the relationship between an agent and its environment. This abstract concept has been interpreted variously across different studies. One of the most recognized definitions within the field identifies affordances with contact heatmaps that highlight interaction contact points. Key research efforts have explored this concept across various dimensions.

Cross-view-AG Luo et al. (2022) introduces a method to learn and apply affordance grounding from diverse human-object interactions to egocentric views, using the new AGD20K dataset. Then, Afformer Chen et al. (2023) extends affordance from image to video by grounding human affordances from demonstration videos to target images through a transformer model. VRB Bahl et al. (2023) presents a method to learn visual affordances from human interaction videos for deployment across various robotic platforms to perform real-world tasks effectively. MIRA Lin Yen-Chen (2022) uses mental imagery and affordance prediction to optimize robot actions in complex scenarios through novel view synthesis. Finally, Affordance Diffusion Ye et al. (2023) develops a method to synthesize hand-object interactions from RGB images, employing a two-step generative approach to enhance affordance prediction capabilities.

Another point worthy of mention is that there is extensive research related to affordance, it often lacks task-specific datasets that bridge to real world scenario. Typically, studies create datasets on a task-by-task basis or borrow from video understanding datasets, which frequently results in a lack of quantitative measures and universal criteria. To address these challenges, SceneFun Delitzas et al. (2024) introduces a fine-grained 3D affordance dataset. Specifically, it offers comprehensive annotations including functional interactive elements, affordance categories, natural language task descriptions, and 3D motion estimation parameters, enabling more rigorous and nuanced analyses.

## 2.2 OBJECT REPRESENTATION FOR ROBOT MANIPULATION

In robot manipulation, point clouds have been effectively utilized in PartManip Geng et al. (2023), leveraging data directly from active sensors for intuitive interactions. Recent advancements, such as those in Weng et al. (2023), employ neural implicit representations like signed distance fields, which are crucial for accurately computing distances to objects, thereby facilitating key robotic tasks. Additionally, Neural Descriptor Fields (NDFs) introduced in Simeonov et al. (2022) enable robots to learn and generalize object manipulation tasks with translational invariance, using self-supervised, SE(3)-equivariant mappings that convert 3D spatial coordinates into meaningful descriptors for interaction.

Further advancing these representations, innovative research, exemplified by F3RM Shen et al. (2023) and LERF-TOGO Rashid et al. (2023), integrates vision-language model embeddings with physical object interactions, allowing robots to determine object positions precisely via language prompts, a pivotal skill for effective manipulation and grasping. Furthermore, the concept of affordances, often visualized as heatmaps on objects, as demonstrated in studies like VRB Bahl et al. (2023), significantly enhances a robot's ability to comprehend an object's functional uses.

## 2.3 ROBOT LEARNING FROM VIDEO

Learning from demonstration has long been an effective strategy for training robots. This approach is exemplified by works such as Liu et al. (2018). However, prior to the development of extensive pretraining techniques, the challenge of training robots to generalize from video demonstrations prompted researchers to pivot towards reinforcement learning. Numerous strategies have been developed to utilize video as a foundational resource. One notable method is imitation from observation Liu et al. (2018), which enables robots to learn basic tasks like sweeping and pushing by emulating human actions from videos, employing context translation and deep reinforcement learning. Additionally, DexMV Qin et al. (2021) offers a robust platform and pipeline for dexterous manipulation using human videos, allowing robots to master complex manipulation tasks by transforming human video demonstrations into robot-executable demonstrations via an innovative translation technique. Lastly, SWIM Mendonca et al. (2023) harnesses large-scale internet data to train world models for robotics, utilizing structured action spaces derived from human videos.

## 3 METHOD

Follow from the work of VRB Bahl et al. (2023), we define an affordance as the contact point and the post-contact trajectory. Our objective is to extend this concept from 2D to 3D. To achieve this, we aim to reconstruct the scene, bind 2D heatmap on to the 3D mesh, and model the post-contact trajectories in 3D using video and camera pose ground truth as inputs.

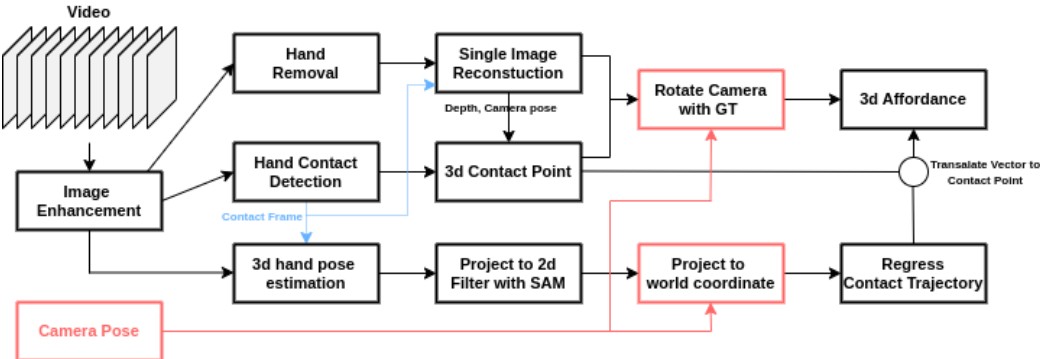

Figure 2: Our method comprises three primary branches. At the top, we employ Dust3r for single image reconstruction. In the middle branch, we project 2D contact points onto the 3D mesh using depth and camera pose information derived from Dust3r. The lowest branch involves regressing the 3D hand pose to obtain the post-contact trajectory. Ultimately, we concatenate these components to derive the 3D affordance.

## 3.1 Contact Detection and Hand Segmentation

To address the motion blur and poor image quality inherent in egocentric videos, we first employ NAFNet Chen et al. (2022) to enhance the quality of the video frames, thereby improving the effectiveness of our subsequent model.

Follow from VRB Bahl et al. (2023), we use off-the-shelf hand contact detector Shan et al. (2020) to detect hand contact of the object in the scene. Additionally, we utilize LangSAM Kirillov et al. (2023) for hand segmentation. For the final step in our process, we use LaMa Suvorov et al. (2021) to remove the hand from images, thereby preparing them for reconstruction.

## 3.2 Single Image Reconstruction

To address the challenges of video demonstrations that lack sufficient imagery of the scene and the contact object, we consider the limitations of traditional 3D reconstruction methods like COLMAP. These methods are often slow and susceptible to failure, typically due to poor image quality, inadequate correspondence between images, or simply not having enough images.

We adopt the 3D reconstruction foundation model, DUST3R Wang et al. (2024). Dust3R is capable of generating point maps and meshes from sparse, low-quality images and does not require detailed camera parameters, making it ideal for scenarios involving sub-optimal image data.

For our process, we input the hand-removed contact image into Dust3R for single-image reconstruction.

## 3.3 3D contact point

We project the 2D contact point, obtained from the contact detection Shan et al. (2020) output, onto the 3D mesh using the camera's intrinsic and extrinsic parameters and the depth data from Dust3r Wang et al. (2024) output. To accurately locate the nearest point on the mesh, we employ a KD tree search to get the contact position. Finally, we bind a heatmap to this 3D contact point to determine the 3D affordance.

## 3.4 Regress Post Contact Trajectory

To determine the post-contact trajectory, we initially employ the HaMeR 3D pose estimation model Pavlakos et al. (2024). To correct inaccuracies in the 3D hand pose predictions, we project these poses back into 2D and refine them using the hand mask from LangSAM Kirillov et al. (2023). These refined poses are then converted back into the world coordinate system using the ground truth camera pose. Finally, we use linear regression to establish the post-contact trajectory based on all

the 3D contact points, after which we calculate the vector and extend the trajectory from the 3D contact point.

# 4 EXPERIMENTS

## 4.1 EXPERIMENT SETUP AND DATASET

In our research, we utilize the Ego4D-exo dataset Grauman et al. (2024). This dataset encompasses a range of actions including cooking, basketball shooting, rock climbing, and piano playing, among others. Predominantly, we have concentrated on the cooking segment for our analysis of 3D affordances. This segment is particularly rich in functional interactive elements such as drawers and refrigerators, as well as tools like knives and bottles that are essential for daily kitchen tasks. These elements provide a comprehensive context for studying and modeling human-object interactions within a 3D space.

In the cooking scene, although numerous scenes appear to be present in ego4d-exo, many of them are duplicative. We identified and categorized the activities into seven principal types: picking up objects, using a knife, peeling vegetables, opening the refrigerator, opening drawers, opening cabinets, and handling a tea bottle. To ensure a thorough evaluation, we manually labeled 2D heatmaps and the ground truth trajectories.

## 4.2 QUALITATIVE RESULTS

We compare the training-free replicated version of VRB Bahl et al. (2023) with our model to demonstrate the effectiveness of our work shown in 3. Our work primarily extends 3D affordance from video to 3D to address the issues of depth, camera pose, and absolute pose in 2D affordance.

More results and failure cases are shown in the appendix.

## 4.3 QUANTITATIVE RESULTS

We evaluate our results using three types of metrics: the completion rate of the 3D affordance, the Intersection over Union (IOU) and Area Under the Curve (AUC) of the 2D heatmap, and the angle between the predicted and ground truth vectors. Note that for the last two metrics, we only consider the complete 3D affordance.

### 4.3.1 COMPLETION RATE

| Categories | Refrig | Drawer | Cabinet | Peel | Cut | Tea | Object |
|------------|--------|--------|---------|------|-----|-----|--------|
| CR | 4/5 | 2/5 | 2/5 | 0/5 | 0/5 | 2/5 | 3/5 |

Table 1: The complete rate of 3D affordance

Our model's effectiveness is assessed across seven diverse tasks: picking up small objects, using a knife, peeling vegetables, opening the refrigerator, opening drawers, opening cabinets, and handling a tea bottle. The results are presented in the accompanying table (see Table 3). Our method is training-free, typically resulting in outcomes of either complete success or failure, with every metric showing zero. We define our complete rate simply according to three criteria.

- The completion of the 3D mesh reconstruction
- The consistent position to project the contact point
- The plausibility of the post-contact trajectory

The task involving the refrigerator was found to be the most effective, attributed to the object's substantial size and the simplicity of the surrounding environment. In contrast, tasks involving the use of a knife and peeling vegetables were the most challenging. After removing the hand, the small size of these objects frequently complicates the reconstruction and accurate projection of contact points.

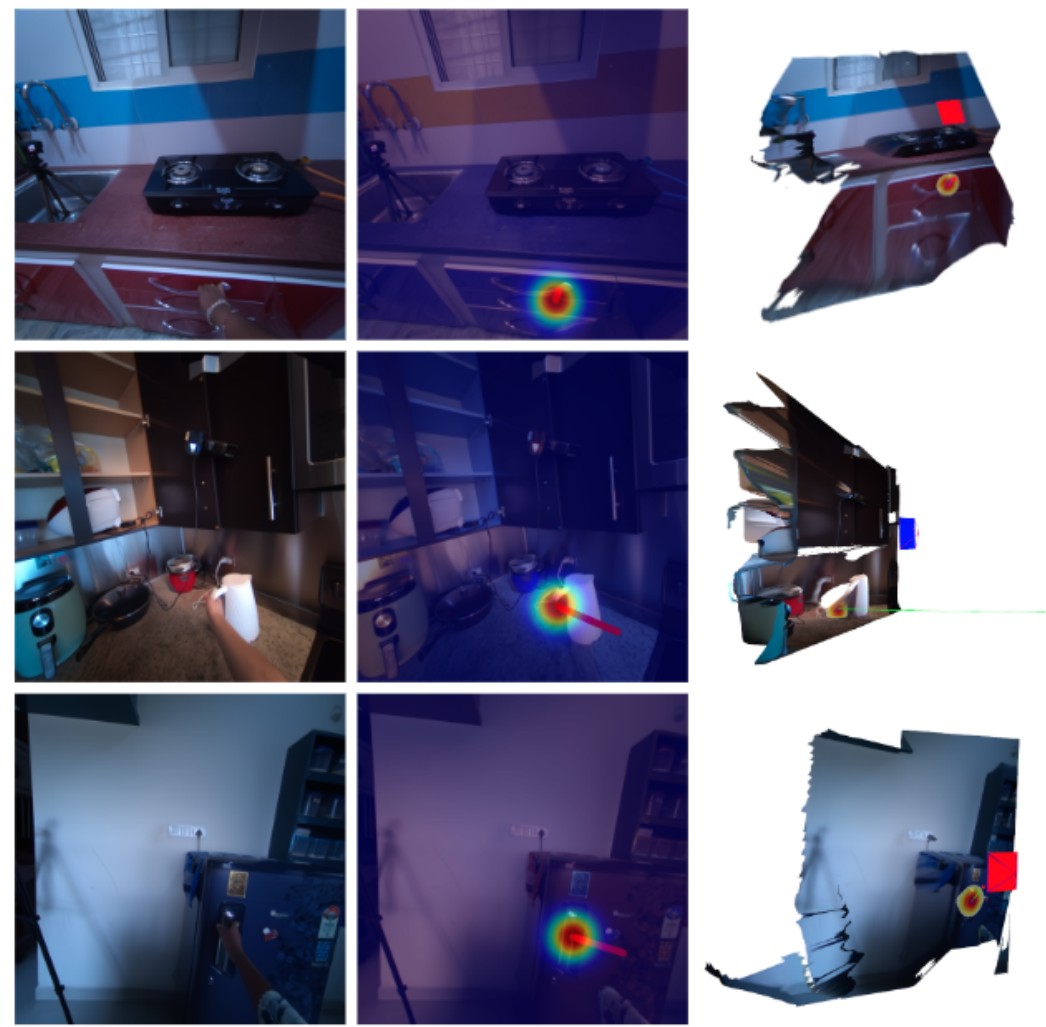

Figure 3: This figure presents the key visualization: the original contact image, a 2D affordance baseline similar to VRB Bahl et al. (2023), and the output from our 3D affordance model. Note: the representations of scale and depth are not intended to be metrically accurate in both the 3D mesh and trajectory.

### 4.3.2 AFFORDANCE HEATMAP

| Categories | Refrig | Drawer | Cabinet | Tea | Object |
|---|---|---|---|---|---|
| IOU | 15.01 | 32.76 | 25.46 | 18.30 | 36.91 |
| AUC | 68.07 | 67.98 | 78.79 | 82.14 | 79.43 |

Table 2: The average 2D heatmap IOU, AUC in percentage with its 2D ground truth.

In this study, we utilize the ego4d-exo dataset, which mainly focuses on video understand in egocentric videos. Thus, we have to manually label 2D heatmaps by ourselves using LabelMe to establish 2D affordance ground truth. To achieve this, we project the corresponding 3D affordance onto the 2D image plane according to its camera pose. Both predicted and actual heatmaps are transformed into binary masks for evaluation metrics such as Intersection over Union (IOU) and Area Under the Curve (AUC), following Li et al. (2024).

The results table 2 indicate that objects like refrigerators perform worse due to their long handles or grasping areas, leading to less accurate heatmap generation. In contrast, smaller objects achieve bet-

ter performance as their contact points closely align with the ground truth, allowing precise heatmap representation.

### 4.3.3 TRAJECTORY ANGLE

| Categories | Refrig | Drawer | Cabinet | Tea | Object |
|---|---|---|---|---|---|
| Angle | 42.7815 | 33.3502 | 36.6735 | 41.9232 | 19.5172 |

Table 3: The average trajectory angle of 3D affordance.

Similarly, to evaluate post-contact trajectories, we have to manually construct ground truth trajectories by selecting two adjacent points on the mesh and computing their cross product to obtain the normal vector at the contact point. We then calculate angle between the predicted and ground truth trajectories and convert this angle into a cosine value for quantitative analysis.

It is important to note that while the normal vector serves as a representative trajectory for categories like tea bottles, some object categories may not have a single, definitive trajectory. Additionally, the ground truth construction is based on the object itself, not on the demonstration video. Therefore, even if the model perfectly predicts the trajectory, there will still be an angle between the two vectors.

### 4.4 IMPLEMENTATION DETAILS

#### 4.4.1 PRE-PROCESS VIDEO

Before inputting the video frames into the model, we first calibrate the fisheye egocentric video to a standard 512*512 video. Subsequently, we manually eliminate any irrelevant video segments.

#### 4.4.2 DEALING LEFT, RIGHT HAND PROBLEM

Our current model often struggles in scenarios involving dual-hand interactions, primarily due to limitations in the hand contact detection model Shan et al. (2020) and the 3D hand pose model Pavlakos et al. (2024). Additionally, most state-of-the-art large models face challenges in accurately distinguishing between the left and right hands, often failing to capture such fine-grained details. To address this issue, considering that our task and video predominantly focus on the right hand, we need to use LangSAM Kirillov et al. (2023) and Lama Suvorov et al. (2021) to remove the left hand initially. This approach will prevent incorrect predictions that could lead to overall result failure.

#### 4.4.3 INCREASE ERROR TOLERANCE IN 3D HAND POSE

We have removed outlier detection in 3D hand pose and applied the LangSAM hand detection mask to filter out points. However, to increase the fault tolerance for outlier points, we have repeatedly dilated the hand masks. This ensures that fewer points are left outside. Particularly, when the hand is viewed from the side, the 3D hand pose model Pavlakos et al. (2024) often predicts poses that are slightly out of bounds.

### 4.5 MULTI-VIEW SETUP

If the contact image is configured with a multi-view setup, the quality of the reconstructed 3D mesh can be significantly enhanced by incorporating additional views, as demonstrated by the capabilities of Dust3R Wang et al. (2024). Additionally, Ego4D-exo Grauman et al. (2024) represents a robust multi-view video dataset. The following are examples 4 that illustrate the improved 3D affordance resulting from these enhancements.

## 5 CONCLUSION

In this paper, we introduce a new method for reconstructing 3D affordances from first-person videos using known camera positions. We harness the capabilities of the 3D foundation model, DUST3R Wang et al. (2024), to reconstruct a sparse view of the object. Subsequently, we project the contact

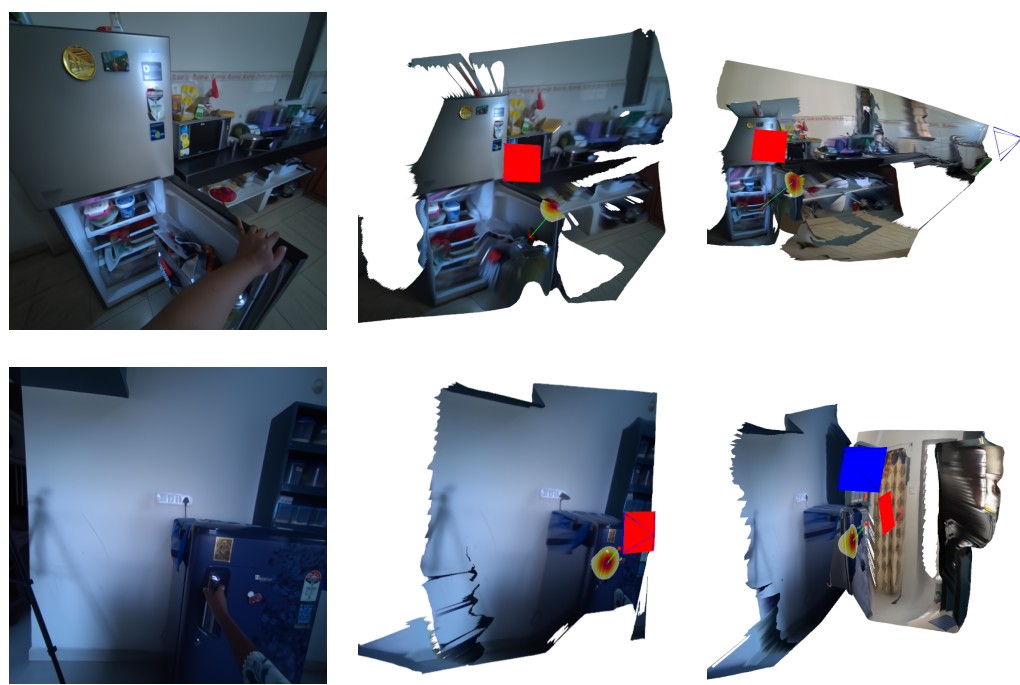

Figure 4: The sequence of images displays the original scenario on the left, a 3D affordance from a single viewpoint in the center, and a 3D affordance from two different viewpoints on the right.

points onto the object mesh and regress these points for post-contact trajectories. This approach allows robots to learn how to use objects by watching human demonstrations without requiring additional training. Our method has the potential to enhance how autonomous robots interact with their environment by utilizing spatial and temporal information.

## 5.1 FUTURE DIRECTION

### 5.1.1 MODEL MORE COMPLEX POST CONTACT TRAJECTORY

Our model currently only uses simple linear regression to model the post-contact trajectory. To address the complex needs of human daily tasks, more advanced models are required to represent features such as curved shapes, circles, or even repetitive actions.

### 5.1.2 HAND OBJECT CONTACT DETECTION

During our work, we found that hand-object contact detection is prone to failure due to the versatility of the objects, contact angles, and lighting conditions. Furthermore, the contact point often cannot be accurately determined through overlap, object detection, or segmentation alone. Additional reasoning is needed to ground accurate location.

### 5.1.3 LEVERAGE DIFFUSION MODEL FOR MORE COMPLETE 3D SCENE

Due to the lack of information in the video, Dust3r Wang et al. (2024) only regresses point-to-point in the 3D point map. Hence, the mesh will lack completeness. Leveraging large generative models for a strong scene prior might be beneficial for completing the 3D scene.

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
