# A APPENDIX

## A.1 THE MAIN PROBLEM OF 2D AFFORDANCE

As demonstrated in Figure 1, if the viewing angle varies, the perceived affordance can appear significantly different, a factor often overlooked by 2D affordance.

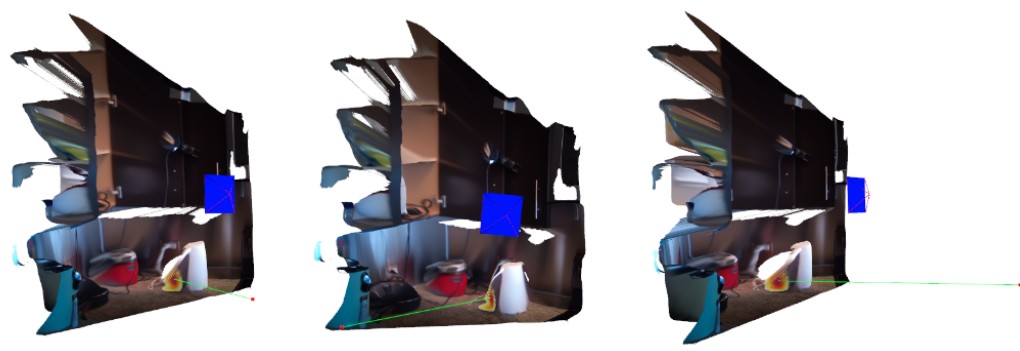

Figure 1: The problem of 2D affordance arises from the real 3D world, as different 2D projections may result according to variations in camera pose and depth

## A.2 MORE 3D AFFORDANCE RESULTS

Figure 2 presents additional 3D affordance results from our model under various situations.

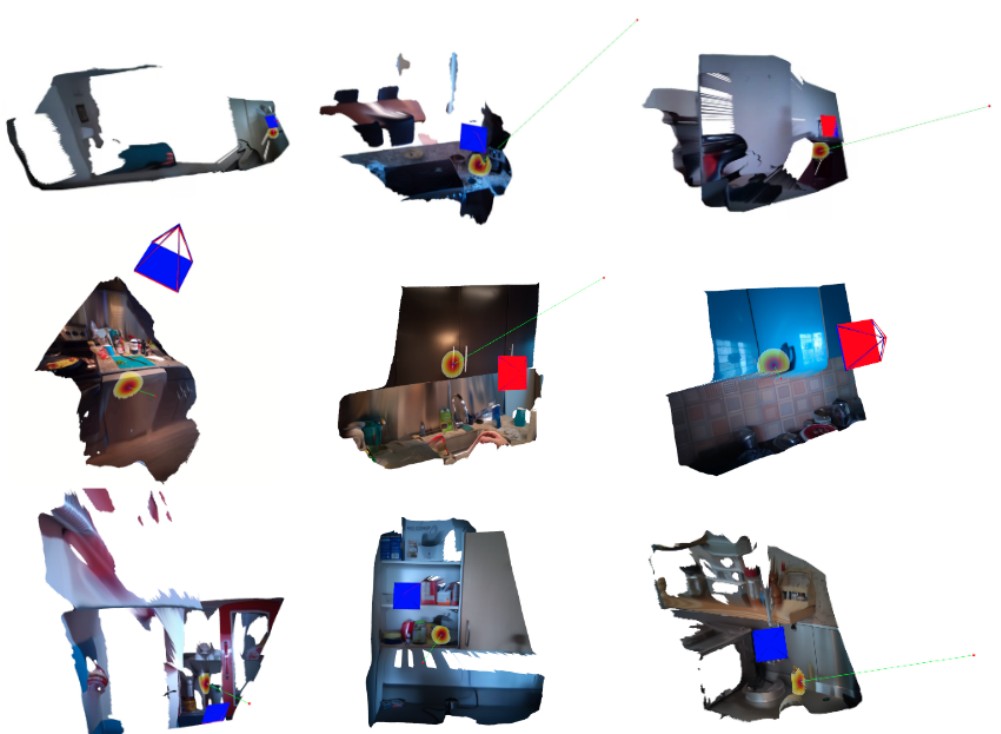

Figure 2: More results of 3d affordance

## A.3 EVALUATION DETAILS

### A.3.1 2D HEATMAP

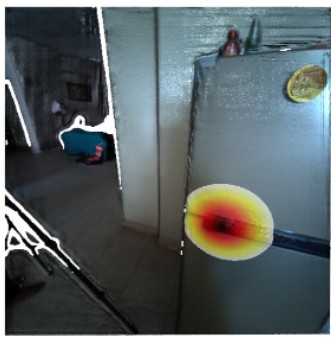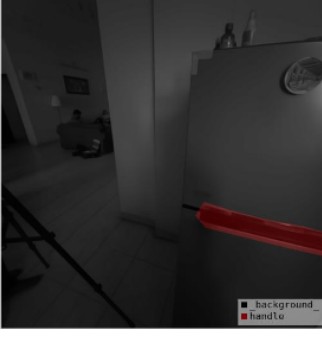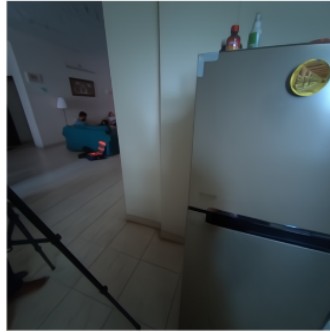

Figure 3: On the left, the predicted heatmap is projected back onto 2D, illustrating potential areas of interest. In the middle, the ground truth is presented, which has been constructed using segmentation labels from LabelMe. On the right, the original image is displayed for reference.

We project the 3D affordance back onto the image using the predicted camera pose. Then, we convert both the predicted and ground truth data into white and black binary masks to calculate the Intersection over Union (IOU) and the Area Under the Curve (AUC).

### A.3.2 TRAJECTORY ANGLE

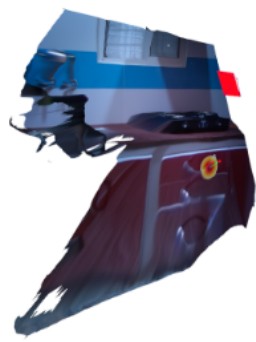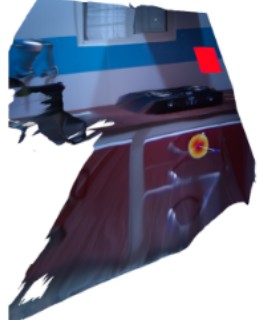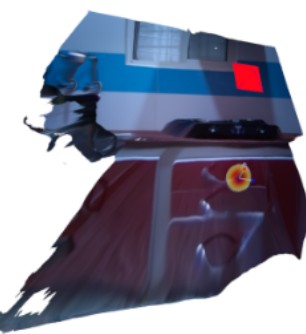

Figure 4: On the left is the predicted trajectory. In the middle is the ground truth, constructed using the cross product of two nearby points. On the right, we overlap the predicted and ground truth trajectories to calculate the angle between vectors.

To determine the ground truth trajectory on a given mesh, we manually select two nearby points to the predicted contact point and compute the cross product of the vectors. Subsequently, we use the normal vector to calculate the angle $\theta$ between the predicted vector $\mathbf{a}$ and the ground truth vector $\mathbf{b}$. This calculation is performed in a Euclidean space using the straightforward formula for the angle between two vectors:

$$\cos\theta = \frac{\mathbf{a} \cdot \mathbf{b}}{\|\mathbf{a}\|\|\mathbf{b}\|}$$

## A.4 FAILURE CASE ANALYSIS

### A.4.1 CONTACT DETECTION FAILURE

Most failures arose from contact detection issues, primarily because the model is outdated and was not trained on the gigantic datasets that are common today. There are three primary modes of failure

shown in 5. First, detection errors occur when the bounds are exceeded, resulting in projections that do not align with the object 5a. Second, uncommon types of contact can lead to detection failures 5b. Finally, in some cases, the system may detect nearby objects even though there is no actual contact with the hand 5c.

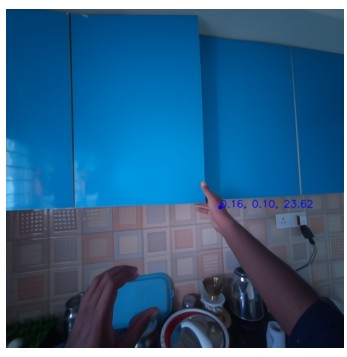
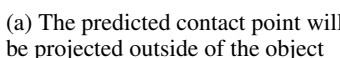
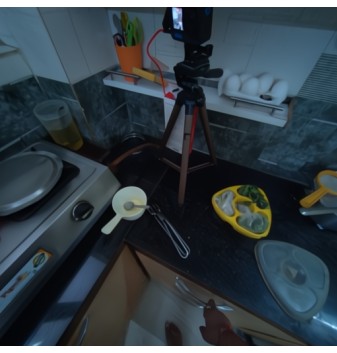
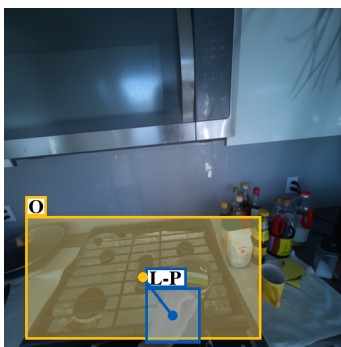

(a) The predicted contact point will be projected outside of the object

(b) Contact in an unusual form will lead to failure in contact detection

(c) wrong detection of object in surroundings

Figure 5: Common contact detection failure

### A.4.2 TRANSPARENT AND REFLECTIVE OBJECT

Transparency in materials leads directly to the failure of 3D reconstruction processes, as nothing can be reconstructed effectively 6b. Conversely, reflective surfaces frequently disrupt contact detection and 3D pose estimation 6c, ultimately causing failures in the 3D reconstruction model, as illustrated in 6

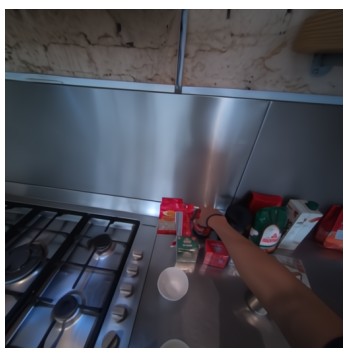
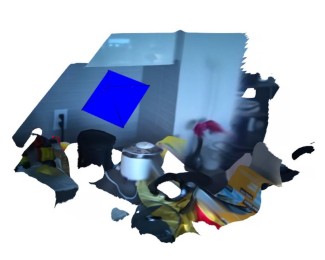
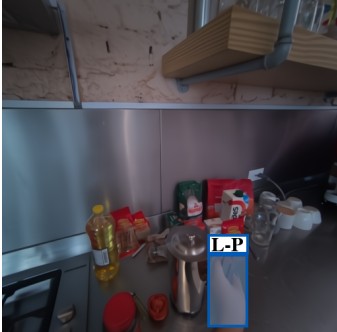

(a) The contact image of a coffee machine

(b) Reconstruction of the coffee machine

(c) The contact detection might failed because of reflective object

Figure 6: Common failure result from transparent or reflective objects

### A.4.3 3D POSE ESTIMATION FAILED

Failures in 3D pose estimation are a frequent issue in our research, often leading to a lack of predictions, as depicted in Figure 7a. These predictions are especially unreliable for lateral views of hands, where there is a notable decrease in accuracy. Additionally, the system may sometimes produce predictions in unexpected locations, exemplified in Figure 7b, where the expected contact point in the middle of the dishwasher is mistakenly predicted on the right side. Furthermore, as depicted in Figure 7c, some hand pose predictions inaccurately appear on the arm instead of on the palm.

### A.4.4 OBJECT TOO SMALL OR TOO COMPLEX SCENE

When picking up objects, it's common to encounter numerous items nearby. Our method mainly identifies the nearest point to the projected 3D point. However, in cluttered areas, as illustrated in

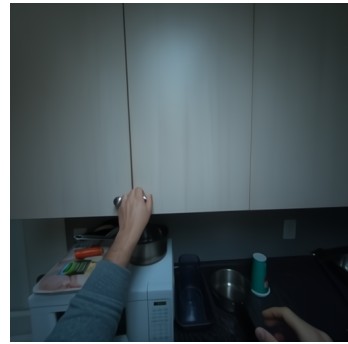 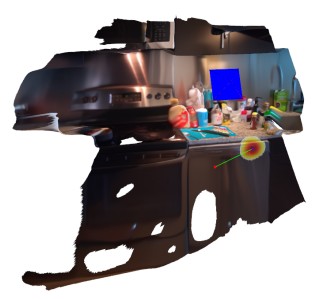 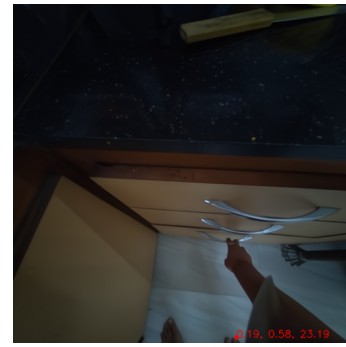

(a) The 3D pose estimation model failed to predict the pose.

(b) The trajectory might have failed because of the incorrect prediction of the 3D hand pose

(c) The 3D hand pose estimation predicts the hand pose on the arm.

Figure 7: 3d pose estimation failed

Figure 8, the projection point may mistakenly align with adjacent objects. In overly complex scenes, the quality of the 3D meshes generated by the reconstruction model tends to decrease. Similarly, in scenes with limited elements, the model might sometimes fail to reconstruct certain parts of the 3D mesh. Additionally, when the object is particularly small, like the handle of a tea bottle shown in Figure 8c, its reconstruction can be less accurate.

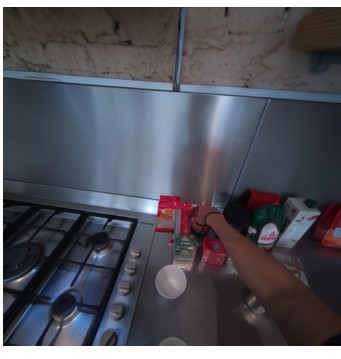 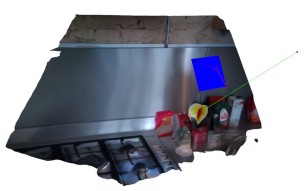 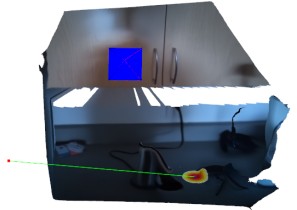

(a) The reference image of the contact image

(b) The projection point is projected onto the nearby object

(c) The tea bottle handle reconstruction failed because of the object is too small

Figure 8: scene problem