# OpenReview forum: "3D Affordance Reconstruction from Egocentric Demonstration Video"
_ICLR.cc/2025/Conference — ICLR 2025 Conference Withdrawn Submission_

### Official Review · Reviewer_MEBn · 2024-10-30

**Soundness:** 2
**Presentation:** 2
**Contribution:** 1
**Rating:** 3
**Confidence:** 2

**Summary:**

The paper presents a method for 3d affordance reconstruction from egocentric videos.  The affordance is defined as the contact point and
the post-contact trajectory. The method uses off-the-shelf hand contact detectors and hand estimators. After the contact point is found, the hand is removed from the image. The hand-removed contact image is reconstructed using  Dust3R. After that, the 3d mesh is obtained using the camera’s intrinsic and extrinsic parameters and the depth data from Dust3R.  This is refined using a 3d pose estimation model and a hand mask given by SAM method. The post-contact trajectory is regressed using the determined 3d contact points.

**Strengths:**

Understanding affordances in 3d -- not only in 2d space is definitely an interesting and significant problem that can help robot action planning.
The fact that that the method is training-free and only requires linear regression could be seen as a strong baseline, showing what could be achieved by stacking of the shelf methods with minimum interventions.

**Weaknesses:**

The paper reads as an application of available off-the-shelf techniques with minimum intervention. Thus it shows a lack of technical novelty.
There is no comparison with other works or baselines and the results are not interpreted.
The metrics used in experiments are not explained properly.

**Questions:**

What does the completion rate of the 3D affordance mean?
I am not sure how to interpret the results, I would have liked to see some baselines or how other works/adaptations of other works would perform.

---

### Official Review · Reviewer_hUGY · 2024-11-01

**Soundness:** 1
**Presentation:** 1
**Contribution:** 2
**Rating:** 3
**Confidence:** 3

**Summary:**

The paper's motivation to extend affordance learning to 3D is both valuable and relevant, addressing a critical need for depth and spatial understanding in robotic manipulation. The authors’ approach, which combines contact detection, hand segmentation, inpainting, and single-view reconstruction, is a straightforward and modular pipeline that provides a clear structure for affordance reconstruction. However, while using single-view reconstruction may simplify the integration of existing methods, it also introduces substantial challenges, particularly in handling occlusions in complex scenarios. The lack of utilization of temporal information or multi-view data limits the method’s ability to address these issues effectively. The experiments are limited in scope and lack the rigor needed to validate the approach, with no substantial comparisons to other methods or benchmarks. Additionally, the presentation and writing quality are somewhat unpolished, with issues in organization and clarity that detract from the paper’s professional appearance. Given these limitations in innovation, depth, and clarity, despite the promising motivation, I recommend a rejection.

**Strengths:**

1. **Meaningful Motivation**: Extending affordance learning from 2D to 3D is indeed a valuable contribution to robotics. By moving beyond 2D, the proposed method captures critical spatial information that can significantly enhance a robot’s ability to understand and interact with objects more effectively. This shift aligns well with practical needs in robot manipulation, where spatial context and distance are essential.
2. **Clear Pipeline for Affordance Detection**: The approach is structured in a straightforward, logical pipeline: it detects contact areas, segments out the hand region, applies inpainting to the hand, uses single-view reconstruction to build the object mesh, and finally projects the detected contact point into 3D. This modular and training-free approach can make the method accessible and adaptable.

**Weaknesses:**

1. **Inconsistent with the 3D Motivation**: Although the motivation emphasizes the importance of 3D information, the method relies on single-view reconstruction, producing meshes without accurate scale. This limitation contradicts the motivation since a lack of true scale prevents the robot from interpreting spatial dimensions accurately. Even though the dataset includes multi-view videos, the method does not leverage these additional perspectives or the temporal data available in video, missing opportunities to refine affordance predictions and ensure correct scaling.
2. **Limited Novelty**: The method decomposes the task into several steps, each handled by existing techniques, aiming for an overall training-free outcome. However, no new solutions are proposed to address the specific challenges of these steps. Single-frame limitations severely impact the accuracy of contact detection, hand segmentation, and inpainting, especially in complex scenes. Employing multi-view or temporal information could help refine these elements, but the paper does not explore such improvements, thus missing potential advancements in these critical areas.
3. **Weak Experimental Setup**: Experiments are limited to a few elements in the kitchen scene from the ego4d-exo dataset, and there is no comparison with other methods. The paper lacks detailed explanations of the dataset annotations, metrics used, or baseline performance, which weakens the validity of the results and the overall persuasiveness of the method.
4. **Writing and Presentation**: The writing quality and presentation are underwhelming. The figures and tables are roughly constructed, the structure is disorganized, and there are numerous typos, which detract from the paper’s professionalism and clarity. Additionally, the content could be more substantial to strengthen the paper’s argumentation and rigor.

**Questions:**

Q1. Line 22: Could you clarify the use of Gaussian heatmaps? Shouldn’t this instead involve camera parameters derived from DUST3R?

Q2. Figure 2: Could you further clarify the step "rotate camera with ground truth"?

Q3. Line 205: For single-view cases, how do you derive camera intrinsics and extrinsics from DUST3R?

Q4. Line 215: Since HaMeR is a monocular pose estimation method, the hand pose is unlikely to have the correct scale in real-world coordinates. I suspect using the ground truth camera pose to convert it to world coordinates may not yield accurate results, potentially questioning the validity of the regressed hand trajectory.

Q5. Line 238: What does "training-free replicated version of VRB" entail? Does it refer to open-sourced code and checkpoints?

Q6. Line 262: How do you assess the completion of 3D reconstructed meshes without ground truth 3D mesh data?

Q7. Line 263: What does “consistent position” mean in this context?

Q8. Line 323: You claim that refrigerators with better reconstruction perform worse than smaller objects with poorer reconstruction. Given the likely occlusion and bad reconstruction of small object contact regions, how can they generate reliable heatmaps? More visualizations would help here.

Q9. Line 335: Given the potentially large errors in the estimated mesh, normal calculations may also be significantly inaccurate. How do you sample adjacent points—are they selected around the vertices of the ground truth contact points?

Q10. Section 4.4.2: Does this mean your method is limited to single right-hand scenarios and cannot handle dual-hand cases?

Q11. Section 4.5: How do you fuse contact point results across multiple views when they are available?

Q12. Line 377: You mentioned single-image reconstruction in line 200, but here you refer to sparse views. Could you clarify?

Q13. Line 408: If using a single view, how do you incorporate temporal information?

---

### Official Review · Reviewer_bc6s · 2024-11-02

**Soundness:** 2
**Presentation:** 1
**Contribution:** 1
**Rating:** 3
**Confidence:** 4

**Summary:**

The paper introduces a method that constructs 3D affordances from egocentric videos to improve robot manipulation. Utilizing the 3D model DUST3R, which rebuilds scenes from sparse images, the method bypasses the need for traditional dense capture techniques. Hand detection identifies interaction points, which are projected into 3D using Gaussian heatmaps and analyzed via 3D hand pose estimation and linear regression. Tested on the ego4d-exo dataset for cooking scenarios, the approach demonstrates enhancements in robotic understanding of dynamic human-object interactions.

**Strengths:**

1. The paper’s focus on affordance is highly relevant for advancing future embodied robotics.
2. The pipeline proposed by the authors is well-structured, utilizing mostly plug-and-play models without requiring additional overhead.
3. The authors validate the feasibility of their pipeline using the Ego4D-exo dataset, demonstrating that the approach is viable.

**Weaknesses:**

1. The writing quality is moderate and would benefit from substantial improvement before final submission. The current version resembles a technical feasibility report more than a polished academic paper. Additionally, the figures in the paper are of relatively low quality.
2. The experimental section lacks detailed comparative results, and no ablation study has been conducted to analyze the proposed pipeline’s components.
3. Adding video demonstrations in the supplementary material could provide valuable insight into the results.

**Questions:**

I believe the current draft appears to be an incomplete version. The authors may consider enhancing their paper by refining the experimental design, providing a more thorough explanation of the method, and improving the quality of the figures.

---

### Official Review · Reviewer_DsVA · 2024-11-04

**Soundness:** 1
**Presentation:** 1
**Contribution:** 2
**Rating:** 3
**Confidence:** 4

**Summary:**

The authors propose lifting the affordance (contact hotspot & post-contact trajectory) prediction problem introduced in "Affordances from Human Videos as a Versatile Representation for Robotics" (Bahl et al., CVPR '23) to 3D. The authors work with the Ego-Exo4D dataset ("Ego-Exo4D: Understanding Skilled Human Activity from First", Grauman et al., CVPR' 24) and claim to use a training-free method. Off-the-shelf methods are used to reconstruct 3D scenes as pointmaps, reconstruct hand meshes, compensate for strong head motion, find hand masks, inpaint hands from scenes prior to lifting to 3D, and extract hand-object contact points in 2D. The contact points are projected into 3D and converted to Gaussian heatmaps indicating affordance hotspots. An evaluation of the method's IoU for affordance hotspot prediction is provided on a self-labeled subset of Ego-Exo4D.

**Strengths:**

The idea of performing affordance prediction in 3D is interesting.

A challenging dataset with significant head motion and object/action diversity is used.

**Weaknesses:**

No baseline comparisons are provided at all, and a dataset created by the authors is used, with little details provided on its quality. This allows for very little conclusions about the efficacy of the proposed method.

There does not seem to be any model used to perform the inference given the hand-free scene other than the off-the-shelf hand-object interaction detector, and thus the affordance predictions can only be made on already contacted objects, limiting the usefulness of the affordance reconstruction work to creating training pseudo-labeled training sets.

The visualizations are of low quality.

The Ego-Exo4D dataset's name is consistently misspelled as Ego4d-Exo.

How exactly are the contact points with the hand calculated?

It is unclear which models exactly are meant in line 353.

**Questions:**

Please address the unclarities in the "Weaknesses" section.

---

### Note · Authors · 2024-11-16

I have read and agree with the venue's withdrawal policy on behalf of myself and my co-authors.